# Crop Yield Assessment Using Field-Based Data and Crop Models at the Village Level: A Case Study on a Homogeneous Rice Area in Telangana, India

**Roja Mandapati** [1,2]**, Murali Krishna Gumma** [1,*]**, Devender Reddy Metuku** [2]**, Pavan Kumar Bellam** [1]**, Pranay Panjala** [1]**, Sagar Maitra** [2] **and Nagaraju Maila** [1]

1   Geospatial Sciences and Big Data, International Crops Research Institute for the Semi-Arid Tropics, Patancheru, Hyderabad 502324, India; roja.mandapati@icrisat.org (R.M.); pavankumar.bellam@icrisat.org (P.K.B.); pranay.panjala@icrisat.org (P.P.); mailanagaraju1998@gmail.com (N.M.)
2   School of Agriculture, Centurion University of Technology and Management, Odisha 761211, India; devender.reddy@cutm.ac.in (D.R.M.); sagar.maitra@cutm.ac.in (S.M.)
*   Correspondence: muralikrishna.gumma@icrisat.org

**Abstract:** Crop yield estimation has gained importance due to its vital significance for policymakers and decision-makers in enacting schemes, ensuring food security, and assessing crop insurance losses due to biotic and abiotic stress. This research focused on rice yield estimation at the field level in the Karimnagar district of Telangana during 2021 and 2022 by employing the leaf area index (LAI) as the primary criterion for integrating remote sensing technology and crop simulation models. Using Sentinel-2 satellite data, the rice crop was mapped with the help of ground data and machine learning algorithms, attaining an accuracy of 93.04%. Crop management data for the DSSAT tool were collected during the field visits; the model results revealed a 0.80 correlation between observed and predicted yields. Due to its strong correlation with LAI (0.82), the normalized difference vegetation index (NDVI) was selected as the critical element for integration with the model. A spatial LAI map was generated using the linear equation developed between the NDVI and LAI. The relationship between LAI and yield was used to create a spatial yield map. The study's findings show that assimilating remote sensing data with crop models enhances the precision of rice yield prediction for insurance companies and policy- and decision-makers.

**Keywords:** leaf area index (LAI); decision support system for agro technology transfer (DSSAT); NDVI; remote sensing; yield

## 1. Introduction

Agriculture drives India's economy, where 60% of the population depends on agriculture and allied sectors (https://icar.org.in/, accessed on 1 April 2020). The Green Revolution has led India from food famine to self-sufficiency in food grain production, improving its economic position. The extensive diversion of agricultural land to other uses has decreased acreage and limited production. The rapid rise in population growth remarkably increased food demand; meanwhile, climate change is limiting crop production. Hence, the agricultural sector requires focused research attention.

Of India's 328.73 million ha land area, the gross cropped area is 197.05 Mha, of which cereals occupy 51.33%, including 22.30% rice [1]. Rice is a primary staple food, with India having the highest designated agricultural area and production. In India, during 2020–2021, the cultivated area, production, and productivity of rice were approximately 44 Mha, 12.1 million tons, and 4.1 metric tons per hectare, respectively [2]. Rice equates to 41% of the overall food grain production and 35% of the country's food grain area, making it a vital component of the national food and livelihood security system [3].

In Telangana, where rice is a dominant cultivated crop, the development of assured irrigation following the completion of irrigation projects and Mission Kakatiya [4,5] resulted in a marked 29.9% to 50.3% increase in production from 2014–2015 to 2020–2021, with a subsequent 29.9% increase from 2019–2020 (19.3 million tons) to 2021–2021 (25.1 million tons) [6]. This cemented the state's position as a leader in the nation for paddy production. Meanwhile, Karimnagar is known as the Rice Bowl of Telangana due to its wide rice intensification area. The area under paddy in Karimnagar during 2019 and 2020 *Kharif* (wet) and *rabi* (dry) was 0.096 and 0.10 M ha, occupying more than 50% of the total cultivated area under Telangana (i.e., 1.96 and 2.13 M ha during 2020 and 2021) [7].

Annual rice crop yields differ considerably due to varied environmental circumstances. Hence, precise and timely crop yield statistics at national, international, and regional levels are becoming progressively essential to overcome food security worldwide [8]. This will ensure effective cropland management, policymaking, import and export decisions, insurance premium pricing, and long-term agricultural food production [9]. Recently many methods, including empirical formulae, remote sensing, and crop simulation modeling, have been applied to estimate crop yield at different scales. Remote sensing has proven extremely beneficial in monitoring agricultural crop growth as it accurately represents the agricultural sector with high revisit frequency and precision [10]. Several techniques based on machine learning have been employed to map crop extent [11–13]. For instance, Earth Resource Data Analysis System (ERDAS) has been used for supervised classification [14,15]. Crop production varies geographically and temporally within an area, making it crucial to assess the pattern of grain yield spatially over the entire field to inform certain management strategies [16,17]. Variations in the yield depend on many internal and external factors. Spatial maps depicting crop yield have been widely employed to elucidate factors contributing to yield variability within plots [18]. Yield loss can also be assessed based on other factors, such as drought and submergence, using remotely sensed data [19]. However, various vegetation indices should be employed as each provides a distinctive set of wave bands that can be associated with distinct crops and their growth phases [20]. Regional differences in crop extent, crop development phases, and leaf area index (LAI) are vital for spatial yield prediction [21].

Different crop simulation models have been used to analyze the potential effects of climate change (sensitivity analysis) on crop yields in varied regions of France, the United States, India, and other countries [22,23]. The DSSAT (Decision Support System for Agro-Technology Transfer) crop model comprises the Crop Environment Resource Synthesis (CERES) model and Cropping System Model (CSM) and can be utilized for approximately 42 crops. Several studies have shown that CERES rice exhibits high agreement between predicted and observed yields under varied management practices in India [24–26]. CERES-maize and wheat have been used to study the effect of spatial precipitation variability and management practices on yield in the United States [27], India [28], and Canada [29]. For yield gap analysis, the DSSAT model is widely used [30,31] under different scenarios.

Though remote sensing and crop modeling methods have advantages, they are limited by the need for more accurate data throughout the crop season and the lack of available satellite data during cloudy days. Various levels of complexity of remote sensing data integration techniques have been assessed by merely incorporating satellite information into simulation models [32]. The LAI can be defined as the unit leaf area per unit land area and is an essential biophysical parameter utilized in many biophysical crop model analyses to depict crop growth stages [33–35]. The crop model LAI has been integrated with remote sensing-extracted biophysical parameters [36,37]. Meteorological and climate data (surface temperature, rainfall, etc.), soil parameters, and management practices are integrated with spatially explicit remote sensing data, such as slope and vegetation indices (e.g., the normalized difference vegetation index (NDVI)) to model crop growth and yield estimations [24,38]. The NDVI is widely used to estimate crop biomass at different growth stages. Although its values vary throughout the crop cycle, the NDVI is the most reliable indicator of light absorption [39,40]. NDVI values from zero (bare soil) to 1 indicate

cultivated soils, while 0.6–1.0 imply dense vegetation at peak growth stages [21]. Remote sensing data alone or combined with other data can estimate crop yield before harvest and provide crop health status via NDVI and LAI [24]. Remote sensing and crop growth modeling are two distinct technologies to address field and regional agronomic issues [41]. Previous research has revealed a paucity of field-level LAI measurements utilizing a zeptometer at various crop stages and a dearth of studies comparing field-level LAI to model LAI. Moreover, earlier investigations have been performed to estimate rice yield using remote sensing technologies at the block or district level. Accordingly, this study has been designed to evaluate the rice yield at the village level under different farm-level management practices using remote sensing and crop simulation models with LAI as the principal component.

## 2. Materials and Methods

### 2.1. Study Area

Karimnagar, in the northeastern Telangana state, has a latitude and longitude of 18°12′ N, 18°68′ N and 78°94′ E, 79°57′ E, and an average elevation of 300 m above mean sea level. A dry climate, with hot summers and cold winters, prevails in this area, with an average annual rainfall of 898.3 mm. Black and red sandy loam soils are the dominant soils in the district. Rice, maize, and cotton are the major crops grown in the district.

### 2.2. Methodology for Optimizing Ground Data Points

Data was collected from four villages to conduct crop-cutting experiments. Ground data points have been optimized based on criteria, including soil type, rainfall map, and elevation map [42]. Four different types of soil have been recognized (clay, clay skeletal, loamy, and loamy skeletal) in the selected study area. Clay and clay skeletal are the major soils occupying more than 80% of the area, whereas loamy and loam skeletal are minor soils occupying less than 20%. Rainfall details for the last 20 years were collected from the Climate Hazards Group Infrared Precipitation with Station data (CHIRPS). The average rainfall in this area was 890 mm under semi-arid conditions, where it receives most precipitation from southwest monsoons. A significant proportion of the district receives less than 1003–1028 mm of precipitation, while a minority receives > 1061 mm. Based on the digital elevation model (DEM), elevation points for the selected study areas were identified. Most of the study area had an elevation > 292 m. Land use land cover (LULC) classification was incorporated as a criterion as rice has been described as the major crop grown in both seasons [43].

Based on these criteria, data from the wet season of four villages (Rukmapur, Veduru-gattu, Elbaka, and Gangipalle) from different mandals covering entire rice-growing regions of the district (Figure 1) were evaluated. Importantly, these villages had different rainfall levels, soil types, and elevation. A combination of the maps were generated to identify the homogeneity in the villages. Detailed methodology adopted for this study is presented in Figure 2.

### 2.3. Data Used

This study employed multispectral data from the European Copernicus Program's Sentinel-2 satellite constellation (https://sentinel.esa.int/web/sentinel, accessed on 1 June 2021). Sentinel-2 captures optical images at a spectral range of 0.47 to 0.6 μm with a high spatial resolution of 10 m above land and coastal areas, which ensures capture with a high revisit frequency of 5 days. Currently, the mission comprises two satellites, Sentinel-2A and Sentinel-2B. Sentinel-2 satellite delivers high-resolution (optical bands at 10 m) images over a wide swath (290 km) in the optical, near-infrared (NIR), and short-wave infrared (SWIR) electromagnetic spectrum. Monthly maximum NDVI with cloud screening for June to November 2021 were collected using data from Sentinel-2 bands 4 (red wavelength) and 8 (NIR wavelength), a spatial resolution of 10 m (Table 1).

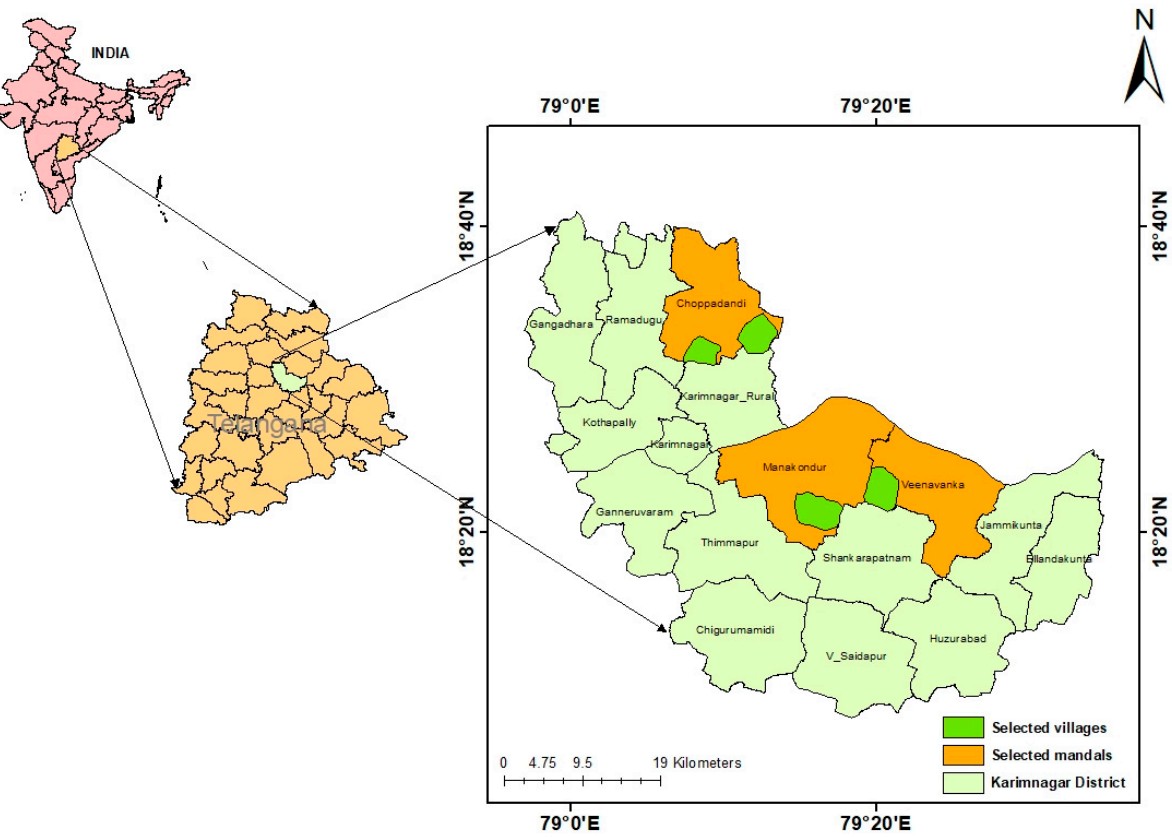

**Figure 1.** Selected study villages in Karimnagar district of the Telangana with location of the study area.

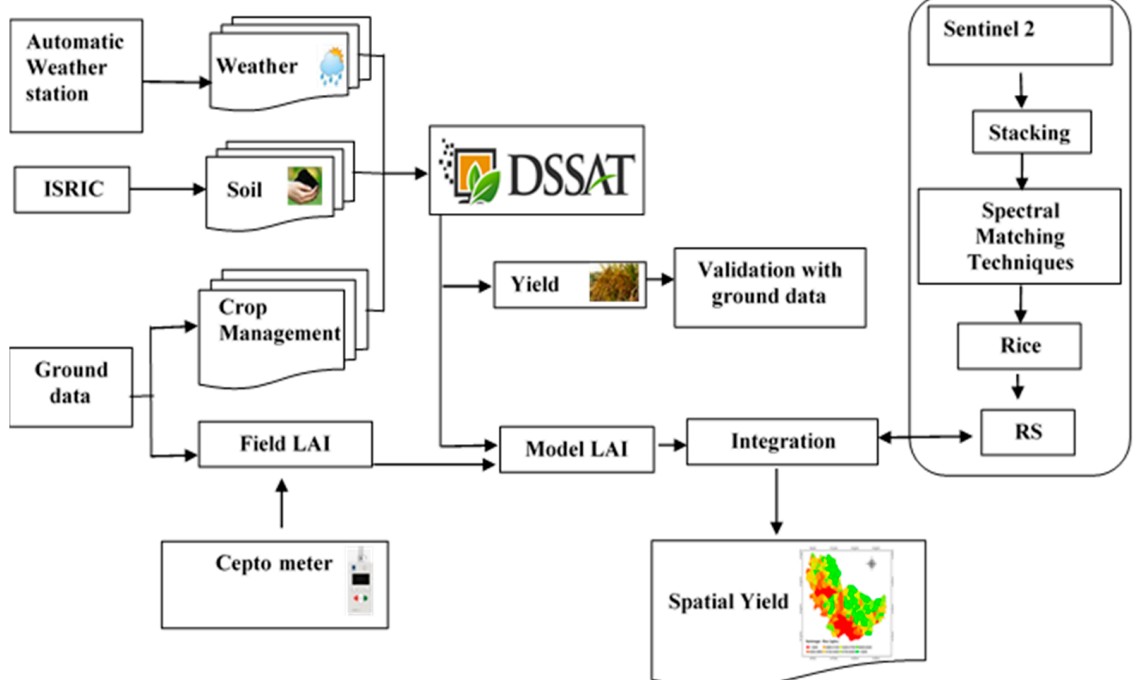

**Figure 2.** Flow diagram representing the methodology for integrating the crop simulation model with remote sensing.

**Table 1.** Details of Sentinel 2.

| Satellite Imagery | Band | Significance |
|---|---|---|
| Sentinel-2 | B4 (Red) | Helps in recognizing the difference between vegetation and other classes |
| | B8 (NIR) | Helps in identifying Water and vegetation |
| NDVI | (B8–B4)/B8 + B4 | Identifies greenness. (vegetation identification) |

Using Sentinel-2, data stacking was performed. Spectral signatures represent crop behavior over time [43,44]. Based on the NDVI spectral signatures, a rice cropland mask was generated, which was used to integrate the LAI for spatial rice yield estimation.

*2.4. Ground Data Collection*

Field visits were scheduled during the rainy season at 20-day intervals to the chosen villages according to the satellite passing time and prevailing weather conditions in the district. With the help of the village agriculture extension officers, farmers' fields with different management practices were selected. Moreover, 220 ground truth points were collected during the wet season, which were used to perform classification and validation.

Fifteen fields were selected in each village with a proper distance from each field to avoid overlapping into a single pixel in the satellite image (Figure 3a). Regular monitoring of the fields was performed. Field IDs were allotted according to the village name for easy identification. Latitude, Longitude, elevation, date of sowing, cultivar details, and quantity of fertilizer applied were some of the major details collected from the farmers. Data collected during the visits are presented in Appendix A. The LAI was estimated using the LP-80 Acicular zeptometer (Figure 3b). In the selected fields, homogenous areas were identified in the crop field where CCEs were conducted on 5 × 5 m plots as per the NSSO Guidelines (Figure 3c). After threshing, the crop biomass weight and grain yield were recorded separately.

*2.5. Crop Model—DSSAT*

DSSAT CERES-Rice (4.7.5 version) was used to estimate the rice yield. CERES-Rice was integrated into the DSSAT-CSM (cropping system model) platform and required meteorological, soil, soil–plant–atmosphere (SPAM), and management modules to simulate crop development, yield, carbon, and water balance [45]. DSSAT crop models require weather, soil, and crop management practices.

2.5.1. Weather Data

Weather inputs, such as daily minimum and maximum temperatures, solar radiation, and rainfall, were necessary for the DSSAT model simulation. For each selected village over the season, meteorological information was acquired from automated weather stations near the villages, except for solar radiation. Solar radiation was computed through the Hargreaves equation [46] using the maximum and minimum temperatures, as follows (Equation (1)):

$$R_s = K_{RS} \sqrt{(T_{max} - T_{min}) \, R_a} \tag{1}$$

where $R_a$ = extra-terrestrial radiation, $T_{max}$ = maximum temperature, $T_{min}$ = minimum temperature, and $K_{RS}$ = adjustment co-efficient (0.16 for interior locations and 0.19 for coastal locations).

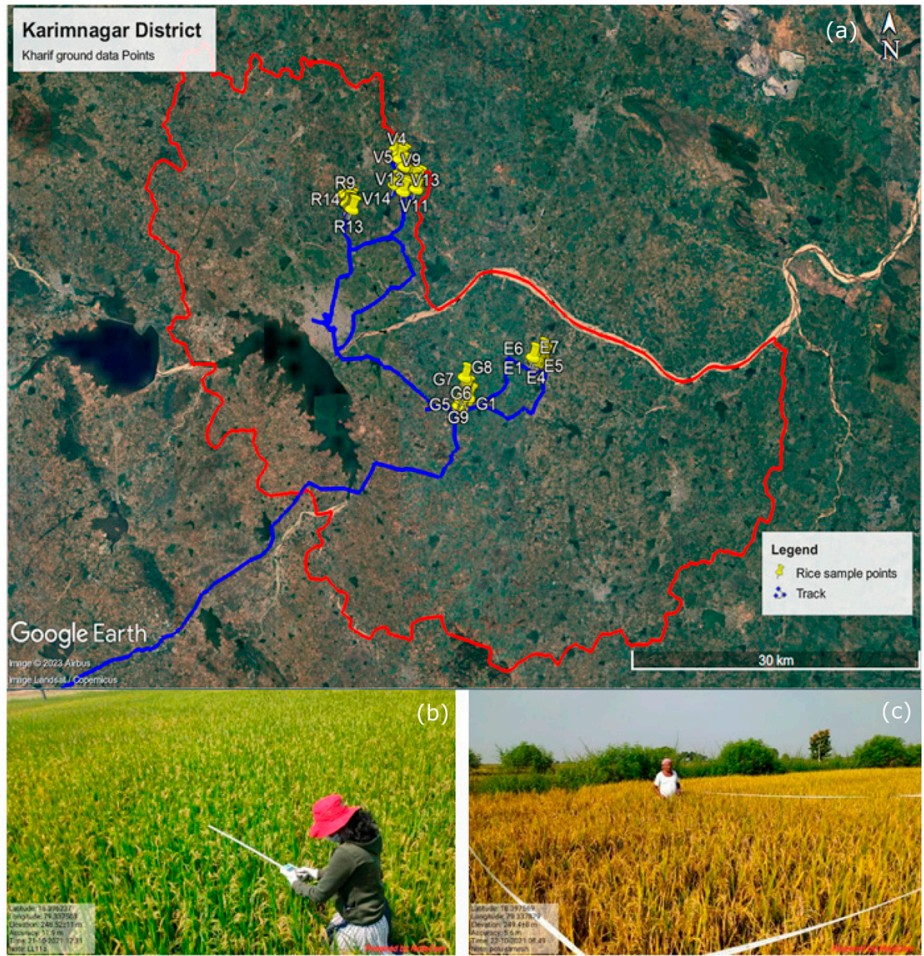

**Figure 3.** (**a**) Ground data sample points collected during Kharif season, (**b**) Estimating LAI using a zeptometer, and (**c**) Conducting crop-cutting experiments.

### 2.5.2. Soil Data

The soil module comprised data from four sub-modules, namely, soil water, temperature, nitrogen, and dynamics, which were updated daily for each soil layer [45]. Soil plant atmosphere (SPAM) analyses root water absorption, potential evapotranspiration (ET), soil evaporation, and plant transpiration from soil, plant, and atmosphere inputs. ET can be estimated using Priestley Taylor or Penman-FAO [47]. DSSAT requires soil parameters, such as soil texture, sand, silt, clay percent, pH, bulk density, EC, cation exchange capacity, and organic carbon percent. These parameters were obtained from ISRIC 2.0 m [48], at a spatial resolution of 250 m. With the use of Google Earth, field-wise data were collected for each field at depths of 0 to 200 cm. Using the general soil database, additional characteristics, including depleted upper and lower limits and saturation limits, were determined based on the soil texture pedo-transfer characteristics provided in the DSSAT models.

### 2.5.3. Crop Management

A questionnaire (Appendix A was developed to collect information from the farmers during field visits to evaluate crop management practices, including cultivar, transplant date, transplant depth, time, and amount of irrigation and fertilizer applied. The most popularly grown cultivars, like MTU1010 (120 days) and BPT5204 (150 days), were selected. A package of practices from individual farmers was collected and fed into the model.

*2.6. Statistical Analysis*

The model's performance was assessed using the coefficient of determination ($R^2$), absolute and normalized root mean square error (RMSE), the Wilmot d index [49], and modeling efficiency (ME). The RMSE and d-index values indicated the model's capacity for predicting experimental data. Following are the equations for measuring model efficacy:

$$R^2 = \left[ \frac{\left[ \sum_{i=1}^{n} (O_i - \overline{O}) \times (S_i - \overline{S}) \right]^2}{\sum_{i=1}^{n} \left[ O_i - \overline{O} \right]^2 \times \sum_{i=1}^{n} \left[ S_i - \overline{S} \right]^2} \right] \tag{2}$$

$$RMSE = \left[ n - 1 \sum_{i=1}^{n} \left( Pi - Oi \right)^2 \right]^{0.5} \tag{3}$$

$$D\ Index = 1 - \left[ \frac{\sum_{i=1}^{n} (P_i - O_i)^2}{\sum_{i=1}^{n} \left[ |P_i'| + |O_i'| \right]^2} \right] \tag{4}$$

$$ME = \left[ \frac{\left[ \sum_{i=1}^{n} (O_i - \overline{O}) - \sum_{i=1}^{n} (P_i - O_i)^2 \right]}{\sum_{i=1}^{n} \left[ O_i - \overline{O} \right]^2} \right] \tag{5}$$

where $S_i$ and $O_i$ represent the predicted and observed values, respectively, n is the number of observations, $\overline{O}$ and $\overline{S}$ are the means of the observed and simulated values, respectively, and $P_i = P_i - M$ and $O_i = O_i - M$ (M is the mean of the observed variable) [50].

## 3. Results

*3.1. Classification*

Supervised classification for the 2021 wet season in the Karimnagar district was performed using the maximum likelihood parametric rule in the ERDAS interface (Figure 4). Rice was the predominant crop throughout the study area, while other crops included cotton and pulses grown under the catchment area, within the southern part of the district.

The total land area used to cultivate rice in the Karimnagar district depicted in the classified map (Figure 4) was 1.43 Lakh ha, compared with the 1.47 Lakh ha reported by the government statistics of Telangana [4]. According to the classified map, the most extensive area was occupied by rice followed by other LULC (0.43 Lakh ha). Water bodies and built-up occupied 0.1 Lakh ha each, while other crops and orchards comprised 0.055 Lakh ha and 0.021 L ha, respectively. An overall accuracy of 93.04% was observed with a kappa coefficient of 0.87 (Table 2).

**Table 2.** Accuracy assessment of the Karimnagar district.

| Classified Data | Producer's Accuracy (%) | User's Accuracy (%) |
|---|---|---|
| Rice | 95.96 | 96.94 |
| Another crop | 85.71 | 85.71 |
| Waterbody | 100 | 100 |
| Orchards | 87.50 | 87.50 |
| Built-up | 85.71 | 100 |
| Other LULC | 83.33 | 71.43 |
| Overall Accuracy | 93.04% | |
| Kappa Coefficient | 0.87 | |

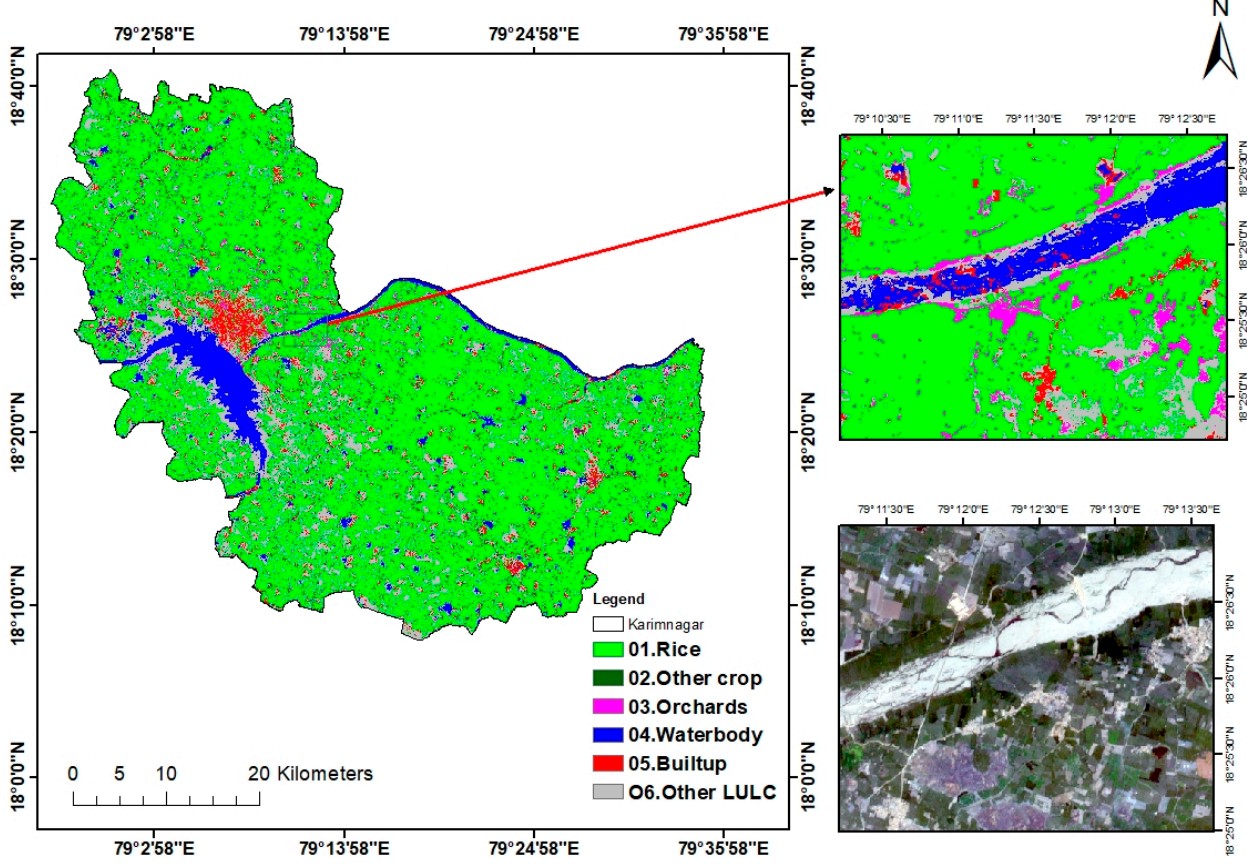

**Figure 4.** Spatial distribution of the rice area in the Karimnagar district during the 2021 wet season.

### 3.2. Model Outputs: Grain Yield

DSSAT CERES Rice was used to simulate grain yield in the Kharif season using weather and soil data. Model-simulated grain yields in each field under varied management practices were compared with the observed yields. Observed yields across the selected villages in Karimnagar ranged from 4100 kg ha$^{-1}$ to 5800 kg ha$^{-1}$, whereas the simulated yields ranged from 4300 kg ha$^{-1}$ to 6000 kg ha$^{-1}$. Among the selected villages, the highest yields were in Vedurugattu (observed: 5200 kg ha$^{-1}$ to 6300 kg ha$^{-1}$; simulated: 4100 kg ha$^{-1}$ to 5800 kg ha$^{-1}$). The model results were statistically analyzed, revealing an overall correlation of 0.80 with an RMSE of 658 kg ha$^{-1}$, D-Index of 0.88, and NSE of 0.70 (Figure 5). From the results, a slight deviation was detected between the observed and simulated yields, where the simulated yields were more than the observed. This was due to the model not considering the loss due to biotic factors, like pests and diseases, and abiotic factors, like lodging, waterlogging, etc.

Individual village-wise statistical analysis was also performed (Table 3). From the statistical results, it is clear that model-simulated yields were reliable and agreed with the observed yields in all selected villages.

**Table 3.** Statistical analysis of the selected villages.

| Village Name | R$^2$ | RMSE | D-Index |
|:---:|:---:|:---:|:---:|
| Elbaka | 0.80 | 374 | 0.86 |
| Gangipalle | 0.87 | 238 | 0.93 |
| Rukmapur | 0.76 | 400 | 0.73 |
| Vedurugattu | 0.72 | 270 | 0.88 |

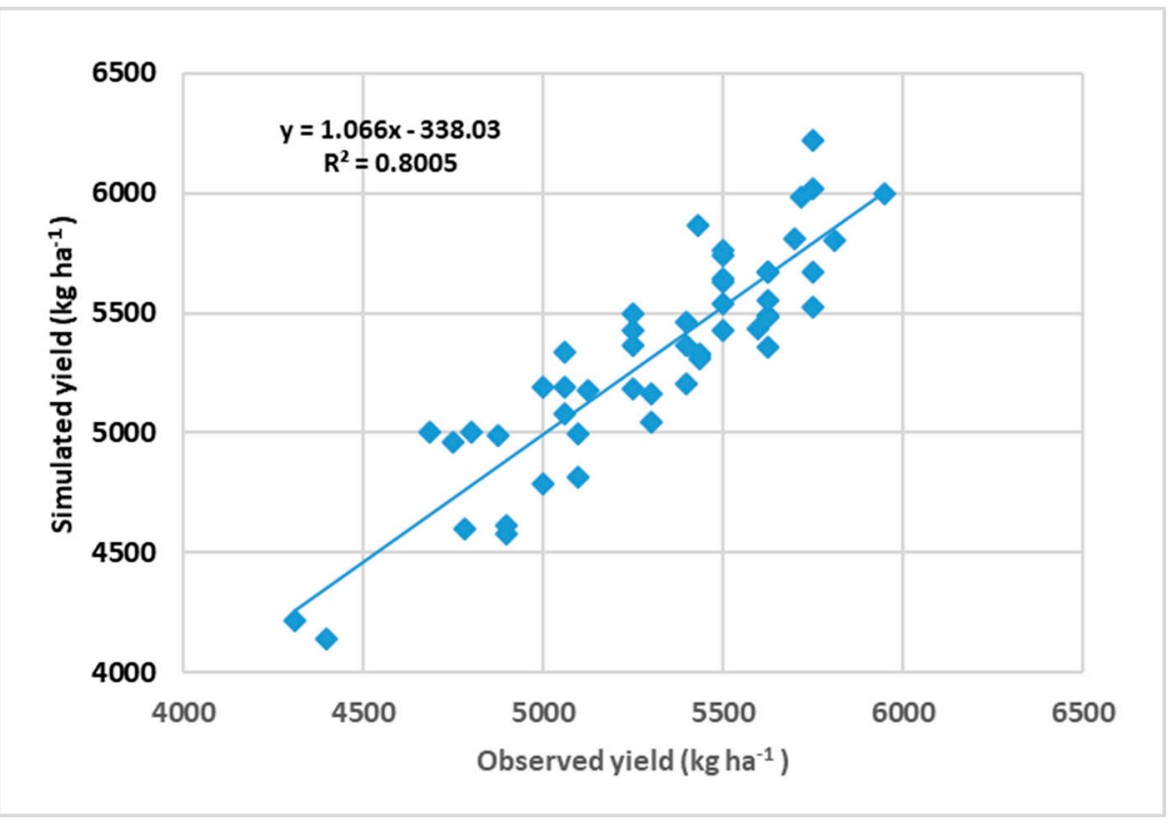

**Figure 5.** Observed and model-simulated rice yields in the Kharif season.

The differences did not lead to a significant decline in the correlation, and the model successfully simulated grain yield during both growing seasons within acceptable limits of error.

### 3.3. Integration of Model LAI and Remote Sensing Product

VV, VH, VV/VH, and NDVI were derived from Sentinel 1 and Sentinel 2 using GEE with filter dates corresponding to ground data visit dates. Among all remote sensing products, NDVI showed a good relationship with LAI ($R^2$ = 0.82; Figure 6); NDVI exhibited the best fit with LAI (correlation = 80%). Hence, it was integrated with the model LAI to estimate spatial yield as it is a commonly used RS index for assessing crop growth and estimating crop yield. Similar findings were previously reported [51,52], revealing that NDVI and LAI show a good correlation of over 0.70 for different plant species.

### 3.4. Generation of Spatial LAI

The corresponding rice pixels were extracted from the classified map to generate spatial *LAI*. First, *NDVI* values at corresponding rice pixels were extracted, and due to the chance of noise in rice pixels, *NDVI* thresholds > 0.4 were considered. To determine RS *LAI*, the linear equation (Equation (6)) generated from the correlation between model *LAI* and *NDVI* was used.

$$LAI = 9.5093 \times NDVI_{max} - 1.8462 \qquad (6)$$

A spatial LAI map was generated using Equation (6). Figure 7 depicts that the maximum LAI was >4.5 in some areas with irrigation and canal facilities.

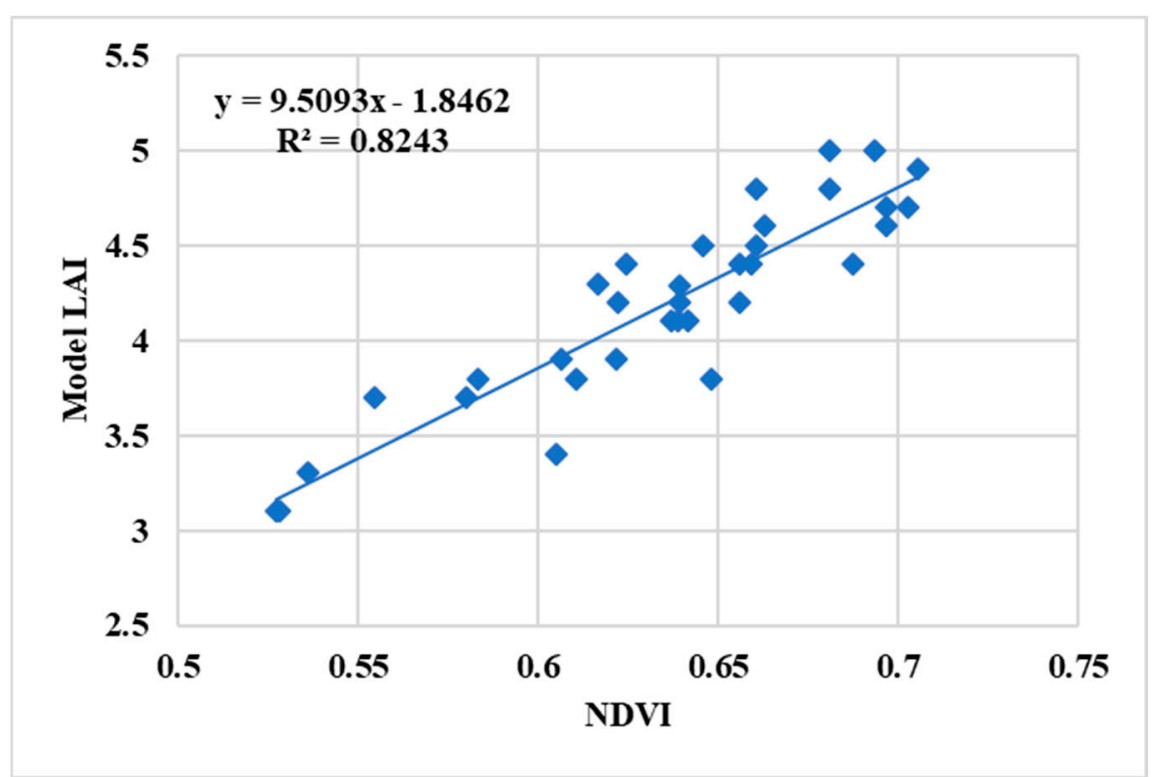

**Figure 6.** Comparison of model simulated LAI with NDVI.

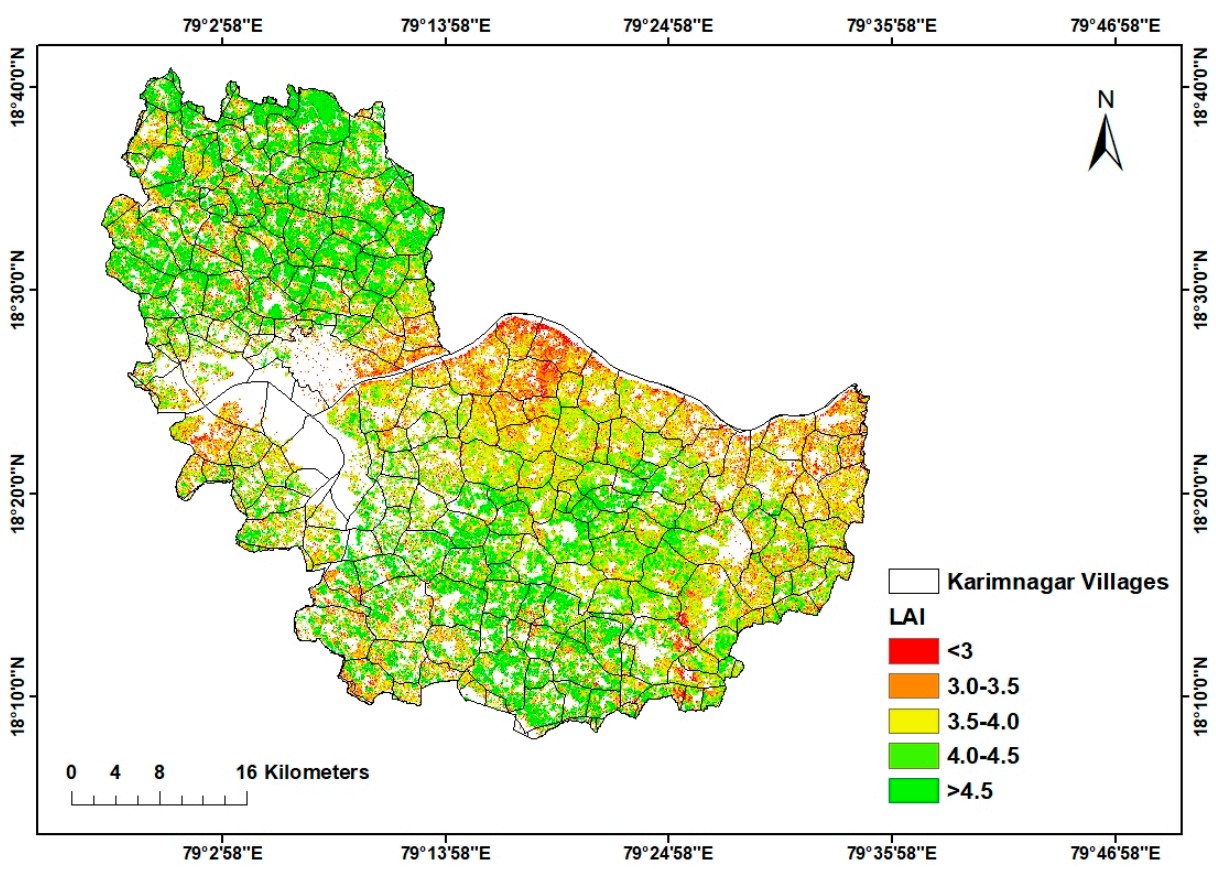

**Figure 7.** Spatial distribution of the LAI map for Karimnagar.

*3.5. Generation of Spatial Rice Yield Map*

A spatial rice yield map was developed using the spatial LAI map, and a linear equation was produced between the model LAI and model yield (Equation (7)), as follows:

$$Yield = 801.89 \, LAI_{max} + 1971.9 \tag{7}$$

A correlation coefficient > 0.80 was obtained between the LAI and model simulated yield. To develop a spatial rice yield map (Figure 8), the linear equation (Equation (6)) resulting from the correlation was applied to the spatial LAI map.

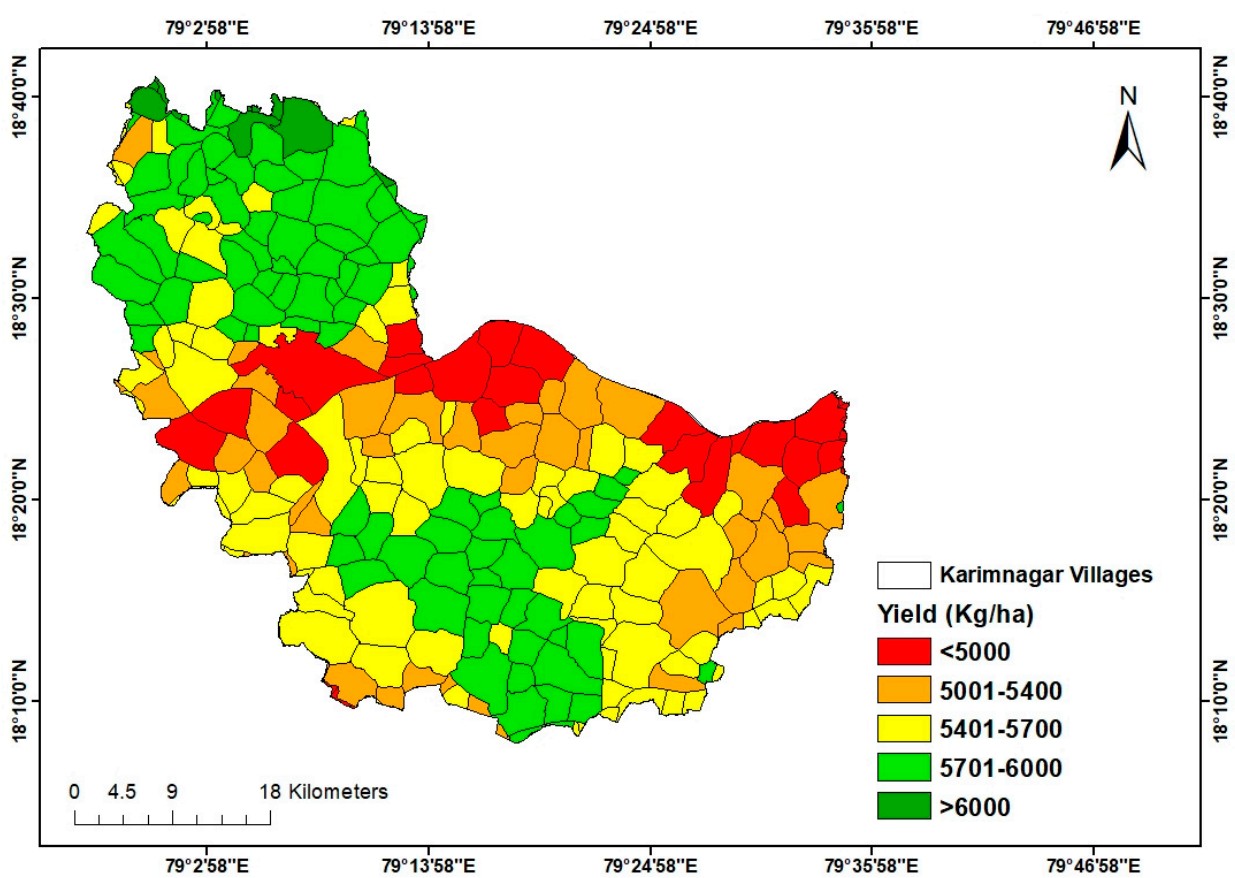

**Figure 8.** Spatial yield map of Karimnagar at the Gram Panchayat (GP) level.

## 4. Discussion

The present investigation emphasizes the integration of crop simulation models with remote sensing techniques to assess rice yield at the village level, with the LAI serving as its core component. The initial step involves mapping the rice-growing regions, which subsequently leads to developing a rice crop mask. The selection of study locations was based on many factors encompassing a wide range of potential combinations. Multiple methodologies have been employed to assess the LAI in diverse agricultural crops, including soybean, maize, potato, wheat, and mustard [37]. The present work employed time series data from the Sentinel-2 satellite to estimate the LAI and utilized the DSSAT agricultural simulation model to enhance the accuracy of crop yield forecasting at the individual field level. The model's simulated LAI was correlated with the remote sensing outputs; nitrogen application was the most significant influencing factor. The necessary weather, soil, and crop data, including the day of transplantation and fertilizer application rates, were obtained and integrated into the crop simulation model. The model and field data exhibited a coefficient of determination ($R^2$) of 0.8 and a Nash-Sutcliffe Efficiency

(NSE) of 0.70. The model established a connection between the simulated LAI and remote sensing outputs, as nitrogen application was identified as the main governing factor of LAI.

The model yielded an average district yield of 5350 kg/ha, comparable to the official statistics of 5014 kg/ha [53]. Based on the observations, it can be asserted that the model exhibits a high degree of reliability in accurately projecting crop yields over a range of different management approaches. Variations in crop output were identified among the villages, which might be attributed to differences in the agricultural management approaches employed in the respective fields. The practice of sowing crops earlier than the optimal period resulted in a decline in agricultural productivity in certain villages, in contrast to the outcomes observed when sowing was carried out at the correct time point. Moreover, applying fertilizers in quantities beyond the recommended dose decreased crop production. The purpose of this study was to demonstrate the significance and essential role of LAI in conjunction with crop models in the evaluation of crop yield assessment. However, relying solely on LAI may not generate accurate estimations since the most notable results have been observed when both LAI and vegetation indices are incorporated into crop models [24,29].

One notable constraint and potential area for improvement identified in this study is the zeptometer's inability to function well when the photosynthetically active radiation (PAR) falls below 400 units. Consequently, this restricts its practicality for data collection under cloudy weather conditions. Utilizing high-resolution data is advisable to achieve a high level of accuracy in the outcomes, given that the farmers possessed small-scale agricultural plots in the villages. The resolution employed in this investigation was set at 10 m. The capacity of DSSAT to address the impact of biotic stress caused by pests, diseases, and weeds is constrained. Hence, integrating dynamic pest and disease models is of utmost importance. It is necessary to calibrate the genotypic coefficients of primary locally cultivated cultivars to ensure their suitability for practical implementation. Adapting the model to fulfill the prescribed criteria presents a formidable task due to its inherent incorporation of intricate protocols and requisite detailed documentation.

This study focused primarily on Sustainable Development Goal (SDG) number 2, pertaining to Zero Hunger and the promotion of sustainable agriculture. The implementation of crop-cutting experiments in a limited area of a field, as opposed to measuring the yield of the entire field, also contributes to the promotion of sustainable agriculture. Collaborative research at the village level in Telangana, India, typically entails forming partnerships among academic institutions, government agencies, and local communities. This approach aligns with the emphasis on partnerships to achieve sustainable development as outlined in SDG 17. Although not the central objective, including crop yield assessment within the climate change framework, can make a valuable contribution to Sustainable Development Goal 13 by addressing climate adaptation and mitigation.

Past studies reporting on the assimilation of remote sensing data in the SIMRIW crop model for rice yield estimation, where COSMO-SkyMed satellite data was used to derive LAI, concluded that SIMRIW-RS has the potential to estimate rice yield accurately (0.80) when LAI of rice is assessed with high accuracy from satellite data [54]. An RMSE of 11.75% was observed between simulated and actual rice yield while integrating remote sensing data in the DSSAT crop model using Taiwan's Particle Swarm Optimization [55]. Moreover, assimilating remotely sensed products with the DSSAT crop model and spectral indices-based regression analysis achieved an $R^2$ of 0.80, NRMSE < 10%, and an agreement of 90%, indicating that these two methods generated superior results to the semi-physical approach [56].

## 5. Conclusions

This study demonstrated the potential of integrating remote sensing products with the crop model by considering LAI as the principal component to estimate the yields at the village level. Optimization of ground data points represented the entire rice growing areas of the district. Mapping of rice growing areas showed an accuracy of 93.04%, comparable

to the government statistics. Input parameters in the DSSAT model were collected during the field visits, and an overall correlation of 0.80, D-Index of 0.88, and NSE of 0.70 were observed, indicating that the model reliably produced results reflective of the field data. NDVI is a remote sensing product that correlated well with LAI ($R^2$ = 82%). Subsequently, a spatial LAI map was generated, which was used to create a spatial yield map. It can be concluded that integrating the remote sensing satellite data with crop models can effectively predict crop yields, which can benefit policy- and decision-makers in implementing insurance schemes.

## 6. Future Line of Work

Models should be properly calibrated to improve the accuracy and consistency of assimilating remote sensing data with crop models. Along with LAI, other variables, including the fraction of absorbed photosynthetically active radiation and soil moisture, can be applied to increase the accuracy in assessing crop yields. As the chosen study area is limited to the district level and includes data from two seasons, the associated data is insufficient to draw definitive conclusions regarding climate change. Hence, this assessment has been designed to evaluate the effects of climate change on crop yield, which can be extended under long-term scenarios.

**Author Contributions:** M.K.G. and R.M. conceptualized the work, drafted the manuscript, and analyzed the results; P.P. and P.K.B. provided technical assistance; P.K.B., R.M. and N.M. collected ground data; D.R.M. and S.M. verified the manuscript. All authors have read and agreed to the published version of the manuscript.

**Funding:** This research received no external funding.

**Informed Consent Statement:** I would like to declare on behalf of my co-authors that the work described was original research that has not been published previously and is not under consideration for publication elsewhere, in whole or in part. All authors have seen a copy of the paper and have approved its submission.

**Data Availability Statement:** Data are contained within the article.

**Acknowledgments:** This research was supported by the ICRISAT. We would like to thank Ismail Mohammed for his support in ground data collection and logistics arrangement. We would like to thank MNCFC for providing secondary information. Our thanks go to ML Jat, Program leader, RF&FS, ICRISAT and the Agronomy department staff from Centurion University for their support.

**Conflicts of Interest:** No conflicts of interest exist regarding the submission of this manuscript, and all authors have approved the manuscript for publication.

## Appendix A

**Table A1.** Questionnaire for crop data collection.

| Farmer details |
|---|
| Name and address |
| Contact no. |
| Location of Plot |
| Area of land holding |
| Previous crop sown |
| Soil type |
| Soil nutrient status |
| Variety name and duration |
| Date of transplanting/sowing |

**Table A1.** *Cont.*

| Irrigation details<br>No. of irrigations<br>Stages of irrigation |
| :---: |
| Fertilizer details<br>Rate of application<br>Stage of application with quantity |
| Organic amendments (if any applied) |
| Pest and disease attack (if any)<br>Name and quantity of insecticides/pesticides used |
| Date of harvesting |
| Yield (Kg/ha) |
| Soil health card details |

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
