# Peer review of "Crop Yield Assessment Using Field-Based Data and Crop Models at the Village Level: A Case Study on a Homogeneous Rice Area in Telangana, India"

_agriengineering, doi:10.3390/agriengineering5040117_

Round 1
Reviewer 1 Report
Crop Yield Assessment Using Field-Based Data and Crop Models at the Village Level: A Case Study on Homogeneous Rice Area in Telangana, India
Very interesting research which is required under the food scarcity of present world. There are some basic issues in this manuscript before accepting it.
1. Title - Homogeneous Rice Area – is not clear
2. Resolution and accuracy are two important cases in NDVI and Satelite images. How do you asses this?
3. How did you use ML concepts?
4. Which particular ML was used?
5. Discussion has to be looked in the context of SDGs.
6. Assess it in the context of climate change which is very important.
7. Please see what has been done in https://doi.org/10.1007/s13201-019-0925-9
8. Please have a small paragraph to compare what others have done in related studies.
Minor
Author Response
I am grateful to you for your thoughtful review of our research, which focuses on crop yield assessment at the village level, particularly within homogeneous rice areas in Telangana, India. Your feedback will help me improve the quality and comprehensiveness of our manuscript. Your comments have prompted me to reconsider enhancing the clarity of our title, aim to provide insights into the resolution and accuracy of our remote sensing data, elaborate on the use of machine learning (ML) concepts, discuss the alignment of our research with Sustainable Development Goals (SDGs), and acknowledge the importance of considering climate change implications. Additionally, we explored the referenced paper and incorporated the learnings from it. It helped us strengthen the clarity and completeness of our research, ensuring it meets the highest standards. For the reader, it ensures that the research is more comprehensible and valuable, contributing to the broader knowledge base on crop yield assessment and its relation to sustainability goals and climate change.

Reviewer 2 Report
The article is very well organized and well written. It is clear about the objective of the work and the technique used. However, minor revisions are requested before publishing the article:
- improve the format of the tables and images because some are outside the margin of the text;
- Revise the formatting of the bibliophraphy because the years in bold are missing;
- In the discussion add some references and some similar work so as to use it as a comparison;
- Add some future perspectives in the conclusions.
Congratulations on a job well done!
Author Response
I sincerely thank for your thoughtful feedback on our manuscript. Your comments highlight opportunities to further enhance the quality and readability of our work. It allows us to refine the manuscript's formatting, bolster its academic rigor through additional references, and offer readers a more comprehensive, informative, and engaging research article.

Reviewer 3 Report
1. Results and discussion needs to be improved in grater sense.
2. As per Figure 2. Flow diagram representing the methodology for integrating crop simulation model with 153 remote sensing, lot of information are not completed in full.
3. The manuscript needs clear presentation with required data.
Moderate editing of English language required
Author Response
I sincerely thank for your constructive feedback on our manuscript. Your comments present an opportunity to enhance the quality and impact of our work. Addressing these comments offers gains from both the writer's and reader's perspectives. It allows us to refine the Results and Discussion sections (4.0) for greater clarity and depth, improve the completeness of our methodology representation, enhance overall presentation for readers, and ensure language correctness.

Round 2
Reviewer 3 Report
Authors have addressed all the points